# Biosynthesis of a sulfated exopolysaccharide, synechan, and bloom formation in the model cyanobacterium *Synechocystis* sp. strain PCC 6803

**Kaisei Maeda[1†], Yukiko Okuda[1], Gen Enomoto[1], Satoru Watanabe[2], Masahiko Ikeuchi[1,3]***

[1]Department of Life Sciences (Biology), Graduate School of Arts and Sciences, University of Tokyo, Tokyo, Japan; [2]Department of Bioscience, Tokyo University of Agriculture, Tokyo, Japan; [3]Faculty of Education and Integrated Arts and Sciences, Waseda University, Tokyo, Japan

**Abstract** Extracellularpolysaccharides of bacteria contribute to biofilm formation, stress tolerance, and infectivity. Cyanobacteria, the oxygenic photoautotrophic bacteria, uniquely produce sulfated extracellular polysaccharides among bacteria to support phototrophic biofilms. In addition, sulfated polysaccharides of cyanobacteria and other organisms have been focused as beneficial biomaterial. However, very little is known about their biosynthesis machinery and function in cyanobacteria. Here, we found that the model cyanobacterium, *Synechocystis* sp. strain PCC 6803, formed bloom-like cell aggregates embedded in sulfated extracellular polysaccharides (designated as synechan) and identified whole set of genes responsible for synechan biosynthesis and its transcriptional regulation, thereby suggesting a model for the synechan biosynthesis apparatus. Because similar genes are found in many cyanobacterial genomes with wide variation, our findings may lead elucidation of various sulfated polysaccharides, their functions, and their potential application in biotechnology.

**\*For correspondence:**
mikeuchi@bio.c.u-tokyo.ac.jp

**Present address:** [†]Department of Bioscience, Tokyo University of Agriculture, Tokyo, Japan

**Competing interests:** The authors declare that no competing interests exist.

**Reviewing editor:** Wolfgang Hess,

## Introduction

Bacterial extracellular polysaccharides establish biofilms for nutrient supply and stress avoidance, and they sometimes support cellular activities such as motility and infectivity (*Woodward and Naismith, 2016*). Generally, the polysaccharide chains consist of a few types of sugars (with or without chemical modifications) and are anchored on cells (capsular polysaccharides, CPS) or exist as non-anchored exopolysaccharides (EPS, also called released polysaccharides). Nonetheless, their molecular structures vary greatly, for example, branching schemes, sugar constituents, and modifications, and thus their physical properties also vary. Bacterial extracellular polysaccharides and lipopolysaccharides are produced and exported via three distinct pathways: the Wzx/Wzy-dependent pathway, ABC-dependent pathway, and synthase-dependent pathway (*Schmid et al., 2015*). Many bacteria can produce several extracellular polysaccharides, and production often depends on environmental conditions. Some extracellular polysaccharides have been appropriated for use as biopolymers for food (i.e. cellulose as nata de coco), food additive (i.e. xanthan as a thickener or an emulsifier), cosmetics (i.e. hyaluronic acid for viscosupplementation), and medicine (i.e. alginate for drug delivery).

Cyanobacteria, the oxygenic photoautotrophic bacteria that inhabit almost every ecosystem on Earth, contribute to the global photosynthetic production (*Flombaum et al., 2013*; *Mangan et al., 2016*). Cyanobacteria produce various extracellular polysaccharides to form colonies, which are planktonic or attached on solid surfaces, likely to stay in a phototrophic niche in nature

**eLife digest** Bacteria are single-cell microorganisms that can form communities called biofilms, which stick to surfaces such as rocks, plants or animals. Biofilms confer protection to bacteria and allow them to colonize new environments. The physical scaffold of biofilms is a viscous matrix made of several molecules, the main one being polysaccharides, complex carbohydrates formed by many monosaccharides (single sugar molecules) joined together.

Cyanobacteria, also known as blue-green algae, are a type of bacteria that produce oxygen and use sunlight as an energy source, just as plants and algae do. Cyanobacteria produce extracellular polysaccharides that contain sulfate groups. These sulfated polysaccharides are also produced by animals and algae but are not common in other bacteria or plants.

One possible role of sulfated, extracellular polysaccharides in cyanobacteria is keeping cells together in the floating aggregates found in cyanobacterial blooms. These are visible discolorations of the water caused by an overgrowth of cyanobacteria that occur in lakes, estuaries and coastal waters. However, little is known about how these polysaccharides are synthesized in cyanobacteria and what their natural role is.

Maeda et al. found a strain of cyanobacteria that formed bloom-like aggregates that were embedded in sulfated extracellular polysaccharides. Using genetic engineering techniques, the researchers identified a set of genes responsible for producing a sulfated extracellular polysaccharide and regulating its levels. They also found that cell aggregates of cyanobacteria can float without having intracellular gas vesicles, which was previously thought to enable blooms to float.

The results of the present study could have applications for human health, since many sulfated polysaccharides have antiviral, antitumor or anti-inflammatory properties, and similar genes are found in many cyanobacteria. In addition, these findings could be useful for controlling toxic cyanobacterial blooms, which are becoming increasingly problematic for society.

(*De Philippis, 1998*). A notable example is the water bloom, a dense population of cyanobacterial cells that floats on the water surface and often produces cyanotoxins and extracellular polysaccharides (*Huisman et al., 2018*). The extracellular polysaccharides are also important for photosynthetic production of cyanobacteria and their application (*Kehr and Dittmann, 2015*; *Kumar et al., 2018*). However, very little is known about their biosynthesis except for extracellular cellulose. A thermophilic cyanobacterium (*Thermosynechococcus vulcanus*) accumulates cellulose to form tight cell aggregation (*Kawano et al., 2011*). This cellulose is produced by cellulose synthase tripartite system unique to cyanobacteria, and its biosynthesis is regulated by temperature and light (*Enomoto et al., 2015*; *Maeda et al., 2018*). In the cyanobacterial genomes, there are still many putative genes for extracellular polysaccharide biosynthesis.

Interestingly, many cyanobacterial extracellular polysaccharides are sulfated, that is, as a sugar modification (*Pereira et al., 2009*). Sulfated polysaccharides are also produced by animals (as glycosaminoglycan in the extracellular matrix such as heparan sulfate) and algae (as cell wall components such as carrageenan) but are scarcely known in other bacteria or plants (*Ghosh et al., 2009*). Major examples of cyanobacterial sulfated polysaccharides are spirulan from *Arthrospira platensis* (vernacular name, 'Spirulina'), sacran from *Aphanothece sacrum* (vernacular name, 'Suizenji-Nori'), and cyanoflan from *Cyanothece* sp. CCY 0110 (*Mota et al., 2020*; *Mouhim et al., 1993*; *Okajima et al., 2008*). These sulfated polysaccharides were found in their biofilms and may be relevant to some ecological functions in cyanobacteria (*Fujishiro et al., 2004*). In addition, the bioactivities (antiviral, antitumor, and anti-inflammatory) of sulfated polysaccharides from cyanobacteria were reported, too (*Flores et al., 2019a*; *Hayashi et al., 1996*; *Ngatu et al., 2012*). However, very little is known about their biosynthesis machinery and physiological functions. On the other hand, biosynthesis and modification of animal sulfated polysaccharides have been extensively studied because of their importance to tissue protection, tissue development, and immunity (*Karamanos et al., 2018*; *Sasisekharan et al., 2006*) and potential applications in healthy foods, biomaterials, and medicines (*Jiao et al., 2011*; *Wardrop and Keeling, 2008*).

The poor understanding about cyanobacterial sulfated polysaccharide biosynthesis is probably due to low or no accumulation of sulfated polysaccharides in typical model species (*Pereira et al., 2009*). More than three decades ago, Panoff et al. reported sulfated polysaccharides in two related model cyanobacteria, *Synechocystis* sp. PCC 6803 and *Synechocystis* sp. PCC 6714 (hereafter Synechocystis 6803 and Synechocystis 6714) (*Panoff et al., 1988*). Recently, Flores et al. confirmed sulfated polysaccharides and reported their enhanced accumulation in a sigma factor *sigF* mutant for global cell surface regulation in Synechocystis 6803 (*Flores et al., 2019b*). In parallel, several papers have studied genes that could be involved in extracellular polysaccharides biosynthesis in Synechocystis 6803, but no clear results were obtained about the sulfated polysaccharide (*Fisher et al., 2013*; *Foster et al., 2009*; *Jittawuttipoka et al., 2013*; *Pereira et al., 2019*). Here, we found that a motile substrain of Synechocystis 6803 showed bloom-like cell aggregation and sulfated EPS production, but a non-motile substrain (a standard substrain for photosynthesis study) did not. By gene disruption and overexpression, we first identified a whole set of genes responsible for sulfated EPS biosynthesis and its regulatory system, opening the way to engineering of their production.

## Results

### Bloom formation and EPS accumulation in *Synechocystis* sp. PCC 6803

We fortuitously found that a motile substrain of Synechocystis 6803 produces EPS and forms floating cell aggregates resembling a typical cyanobacterial bloom. We established a two-step culture regime (2-day bubbling culture and subsequent standing culture without bubbling under continuous light) for reproducible formation of bloom-like aggregates (*Figure 1A,B*). The first (bubbling) step allows for cell propagation and EPS production, whereas the second (standing) step allows for massive cell aggregation and flotation, even though Synechocystis 6803 does not possess genes for intracellular gas vesicles (*Harke et al., 2016*). In Synechocystis 6803, cell flotation accompanying the generation of extracellular gas bubbles was suppressed by inactivation of photosynthesis (*Figure 1C*), suggesting that gas derived from photosynthesis drives the upward movement of cells embedded in viscous EPS. The non-motile, glucose-tolerant substrain—commonly used for photosynthesis research—did not aggregate or float.

We first isolated crude viscous EPS from the bloomed culture by membrane filtration (*Figure 1— figure supplement 1A*). The crude EPS consisted of polysaccharide but very little protein or nucleic acid and its abundance remained unchanged during the second culture step (*Figure 1—figure supplement 1B*). As a common feature of diverse EPS biosynthesis systems in bacteria, membrane-bound glycosyltransferases are particularly important (*Schmid et al., 2015*). In the Synechocystis 6803 genome, 59 genes were annotated as glycosyltransferases. Twelve of them were uncharacterized and predicted to encode transmembrane helices. So, we disrupted five of them initially and found that *slr5054* is essential for bloom formation (*Figure 1D,E* and *Figure 1—figure supplement 2*). The viscous EPS preparation was improved by removing cells before filtration to avoid cell-associated polysaccharides such as CPS (*Figure 1F*). Δ*slr5054* lacked most of the EPS present in the wild type (WT), whereas the CPS and free polysaccharide fractions were similar in the WT and Δ*slr5054* (*Figure 1G*). Then, we performed Alcian blue staining to examine the acidity of the EPS (*Figure 1— figure supplement 3*). Generally, sulfated polysaccharides are stained at pH 0.5 condition, while acidic polysaccharides, which contain sulfate groups and/or carboxylate groups (such as uronic acids and carboxylate modification), are stained at pH 2.5 condition (*Bellezza et al., 2006*). The EPS from WT was clearly stained under both pH conditions, strongly suggestive of the sulfate modification.

Gene cluster for the biosynthesis of viscous EPS *slr5054* resides on a megaplasmid, pSYSM, in a large gene cluster (*sll5042–60*), which we named *xss* (extracellular sulfated polysaccharide biosynthesis) (*Figure 2A*, *xssA–xssS*). This cluster includes two genes for sulfotransferases (*xssA*, *xssE*), eight genes for glycosyltransferases (*xssB*, *xssC*, *xssG*, *xssI*, *xssM*, *xssN*, *xssO*, *xssP*), three genes for the polysaccharide polymerization system (Wzx/flippase; *xssH*, Wzy/polymerase; *xssF*, and polysaccharide co-polymerase [PCP]; *xssK*), one gene for a putative transcriptional regulator (*xssQ*), a pair of genes for the bacterial two-component phosphorelay system (*xssR*, *xssS*), and genes encoding several small proteins of unknown function (*Supplementary file 1*, *Figure 2A*). All genes except those of unknown function were disrupted individually with a read-through cassette, and segregation was confirmed by colony PCR (*Figure 2—figure supplement 1*). Bloom formation and sugar content of

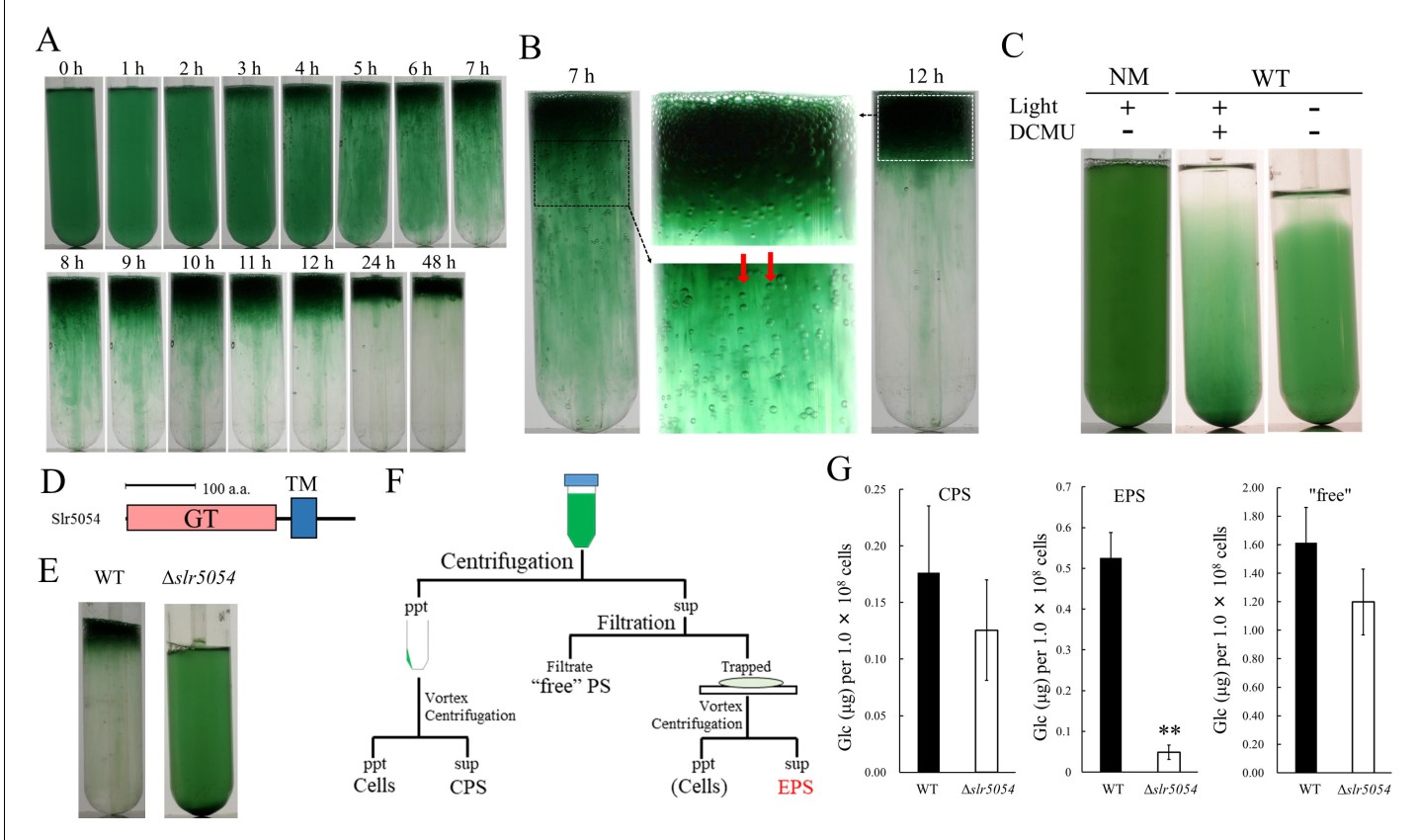

**Figure 1.** Bloom formation and exopolysaccharides (EPS) isolation. (**A**) Time course of bloom formation by wild-type (WT) Synechocystis 6803 during the second step of culture. Extracellular gas bubbles are formed and trapped in viscous EPS (~1 hr). Green vertical columns with bubbles become apparent at 4 hr. Those trapped gas bubbles slowly rise together with the viscous columns. (**B**) Enlarged images showing gas bubbles trapped in EPS. Vertically aligned bubbles are indicated by red arrows. (**C**) Lack of bloom formation in the non-motile substrain (NM) or WT with or without light and the photosynthesis inhibitor DCMU at 48 hr of the second step of culture. (**D**) Domain architecture of Slr5054. GT, glycosyltransferase domain; TM, transmembrane region. (**E**) Lack of bloom formation in Δslr5054 after standing culture for 48 hr. (**F**) Isolation of EPS from the first step of culture. Cells and capsular polysaccharides (CPS) were removed from the culture by centrifugation, and EPS in the supernatant was separated from 'free' polysaccharide (PS) by membrane filtration followed by a second centrifugation to remove residual cells. CPS was collected from the cell pellet after vortexing and centrifugation. (**G**) Sugar content of fractions from WT and Δslr5054. Error bars represent SD (CPS, n = 6; others, n = 3; **p<0.005). The online version of this article includes the following source data and figure supplement(s) for figure 1:

**Source data 1.** Raw data of sugar quantification in wild type (WT) and Δslr5054.
**Figure supplement 1.** Isolation of crude exopolysaccharides (EPS) and sugar analysis.
**Figure supplement 2.** Bloom formation by several glycosyltransferase mutants.
**Figure supplement 3.** Alcian blue staining of isolated exopolysaccharides (EPS) from wild type (WT).

the EPS fraction were reduced in many mutants (*Figure 2B,C*). In particular, bloom formation was completely abolished in ΔxssA, ΔxssB, ΔxssF, ΔxssH, ΔxssK, ΔxssM, ΔxssN, and ΔxssP, in which EPS accumulation was also suppressed. Certain glycosyltransferase mutants (ΔxssC, ΔxssG, ΔxssI, ΔxssO) formed blooms but accumulated little EPS, and neither bloom formation nor EPS accumulation was substantially altered in one sulfotransferase mutant (ΔxssE). In general, the Wzx/Wzy system in bacteria produces various EPS, lipopolysaccharides, and CPS through four steps: (i) biosynthesis of a heterooligosaccharide repeat unit on a lipid linker at the cytoplasmic side of the plasma membrane by a series of glycosyltransferases and modification enzymes, (ii) flip-out of the unit to the periplasmic side by Wzx, (iii) polymerization by transfer of the nascent polysaccharide chain to the repeat unit by Wzy, and (iv) export of the EPS chain through the periplasm and outer membrane via PCP and the outer-membrane polysaccharide export protein (OPX) (*Islam and Lam, 2014*; *Schmid et al., 2015*). It is very likely that the *xss* cluster harbors a whole set of genes for the Wzx/Wzy-dependent pathway except a gene for OPX.

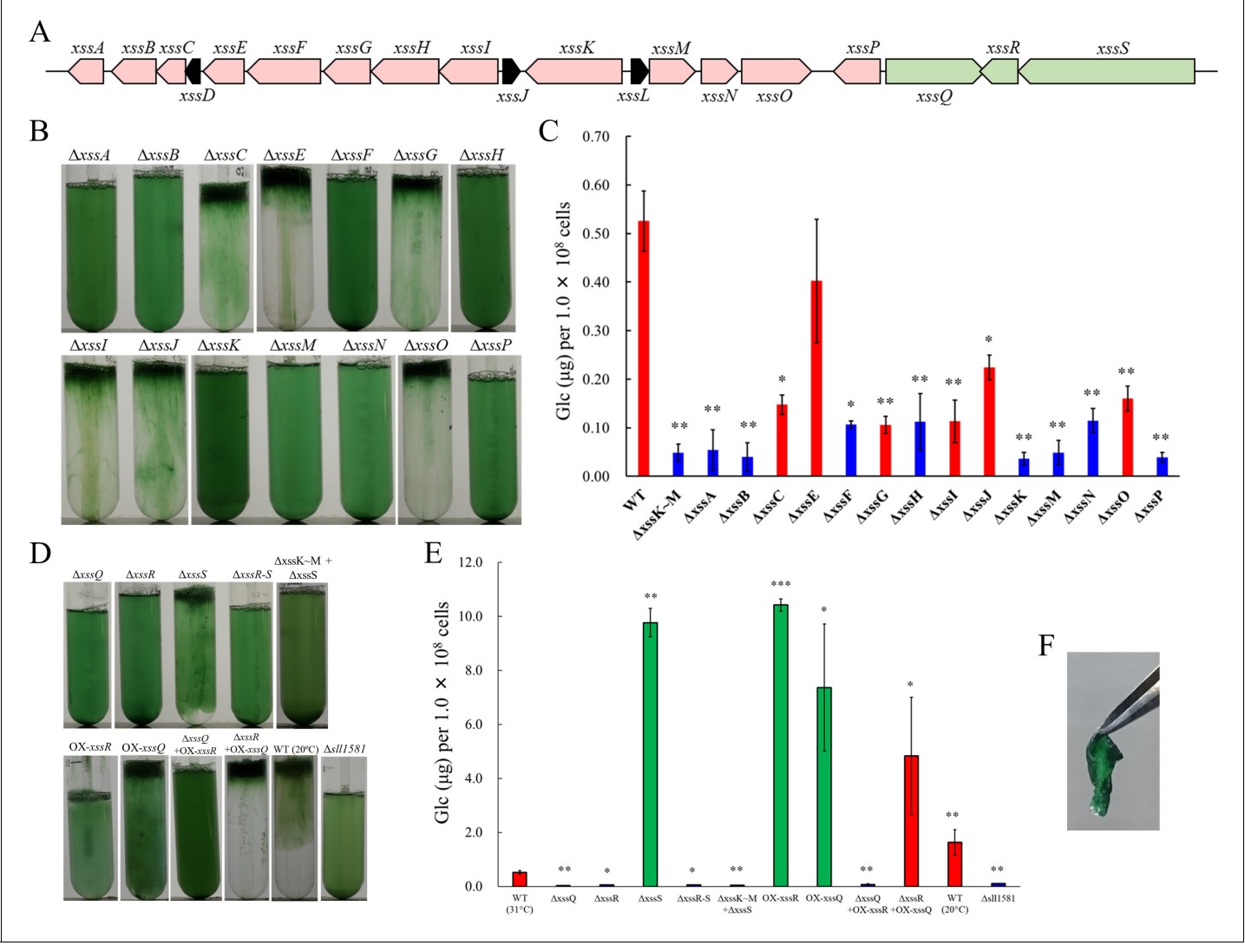

**Figure 2.** The extracellular sulfated polysaccharide biosynthesis (*xss*) gene cluster and phenotype of *xss* mutants. (A) The *xss* gene cluster. Red, polysaccharide biosynthesis genes; green, regulatory genes; black, genes of unknown function. (B) Bloom formation by the mutants carrying disruptions in the polysaccharide biosynthesis *xss* genes. (C) Total sugar content (μg glucose per $1 \times 10^8$ cells) of the exopolysaccharides (EPS) fraction from mutants in b. Red bars, bloom-forming mutants; blue bars, non-bloom-forming mutants. Error bars represent SD (wild type [WT] grown at 20℃, n = 6; others, n = 3). Statistical significance was determined using Welch's *t* test (*p<0.05, **p<0.005, ***p<0.0005). (D) Bloom formation by regulatory mutants, WT grown at 20℃, and outer-membrane polysaccharide export protein (OPX) mutant (Δ*sll1581*). (E) Total sugar content of the EPS fraction from mutants in d. Red bars, bloom-forming mutants; green bars, excess-bloom-forming mutants. (F) A sheet of OX-*xssR* cells was stripped off from the agar plate by tweezers. The culture temperature was 31℃ unless otherwise stated.

The online version of this article includes the following source data and figure supplement(s) for figure 2:

**Source data 1.** Raw data of sugar quantification in wild type (WT) and mutants.

**Figure supplement 1.** Agarose gel electrophoresis to assess segregation of mutants based on PCR data.

**Figure supplement 2.** Domain architecture and proposed signal transduction pathway for XssS/XssR/XssQ.

## Regulation of the sulfated EPS biosynthesis

The sensory histidine kinase mutant Δ*xssS* accumulated a much larger amount of EPS than WT, whereas mutants of the cognate response regulator *xssR* and transcriptional regulator *xssQ* had a null phenotype with regard to both bloom formation and EPS accumulation (*Figure 2D,E*, *Supplementary file 2*). The double mutant Δ*xssS*/Δ*xssR* had a phenotype similar to that of Δ*xssR*. Overexpression of *xssR* or *xssQ* (OX-*xssR*, OX-*xssQ*) resulted in strong bloom formation as well as marked accumulation of viscous EPS, similar to that seen for Δ*xssS*. The combination of *xssQ*

disruption and *xssR* overexpression (Δ*xssQ* + OX *xssR*) abrogated bloom formation and EPS accumulation, whereas the combination of *xssQ* overexpression and *xssR* disruption (Δ*xssR* + OX *xssQ*) resulted in a pronounced phenotype of bloom formation and EPS accumulation. These results suggested that the sensor histidine kinase XssS suppresses the response regulator XssR, leading to activation of the transcriptional activator XssQ. Notably, the OX-*xssR* and OX-*xssQ* strains formed sticky, non-motile, biofilm-like colonies on agar plates that could be picked by tweezers (*Figure 2F*).

XssQ is a new type of the *s*ignal *t*ransduction *A*TPase with *n*umerous *d*omains (STAND) protein, because it harbors an N-terminal helix-turn-helix transcriptional DNA-binding domain (*Figure 2—figure supplement 2*). Typical STAND proteins possess a three-domain module with ATPase activity and are involved in processes such as apoptosis and immunity in animals, plants, and some bacteria (*Danot et al., 2009*). Using real-time quantitative PCR (qPCR), we compared gene expression in the *xss* cluster for WT, Δ*xssS*, and Δ*xssQ* (*Figure 3A*). Expression of five genes (*xssA*, *xssB*, *xssE*, *xssN*, *xssP*) was very low in Δ*xssQ* compared with WT, whereas that of *xssF*, *xssH*, and *xssK* was not substantially affected. These results suggested that XssQ transcriptionally activates genes encoding sulfotransferases and certain glycosyltransferases but not genes for polymerization and export via the Wzx/Wzy system. qPCR analysis of gene expression in Δ*xssS* revealed a tendency for slight up-regulation of *xssB*, *xssE*, *xssN*, and *xssP*. We next performed RNA-seq of WT, Δ*xssS*, and Δ*xssQ* to

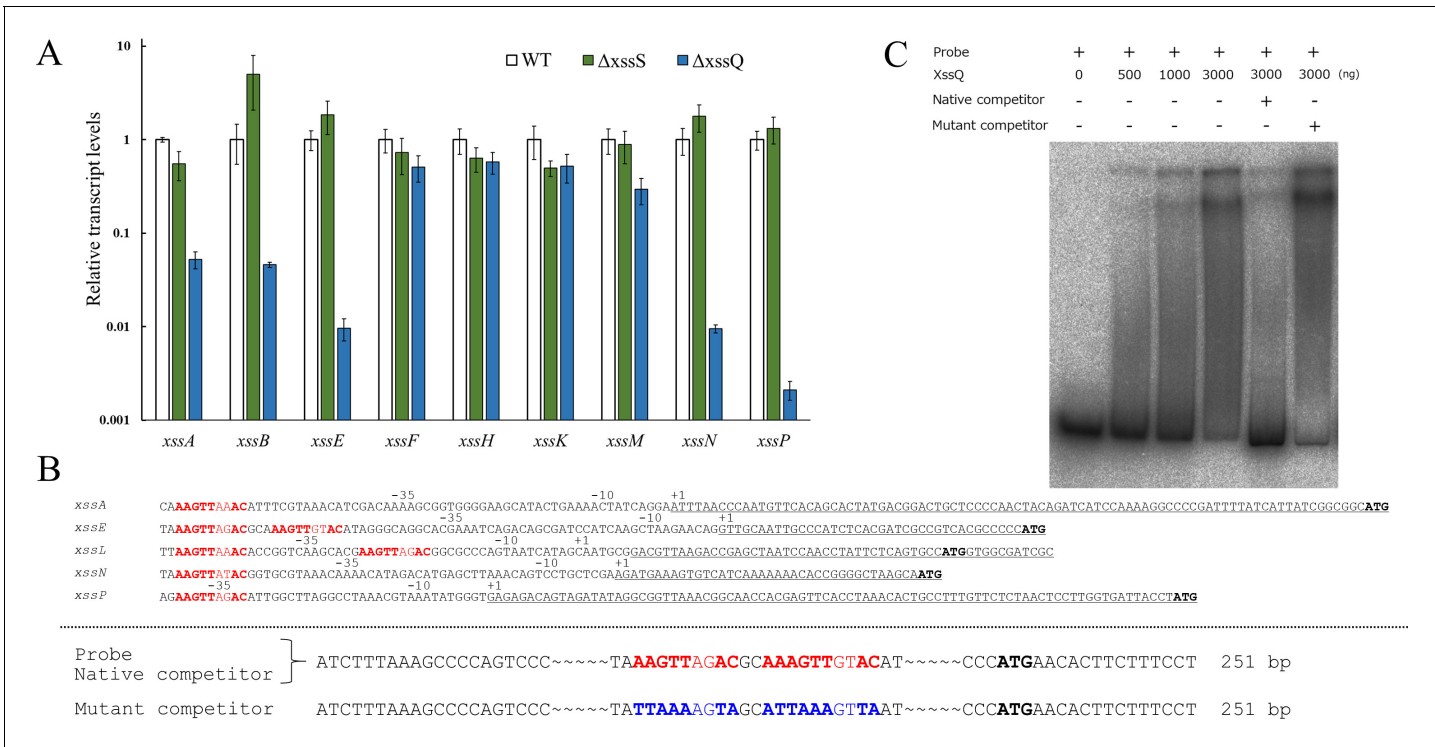

**Figure 3.** Transcriptional regulation of extracellular sulfated polysaccharide biosynthesis (*xss*) genes. (**A**) Transcript levels for *xss* genes in wild type (WT), Δ*xssS*, and Δ*xssQ* measured by quantitative PCR (qPCR). The internal standard was *rnpB*. Relative expression levels were obtained by normalization to the transcript levels of each gene in WT. Error bars represent SD (n = 3, biological triplicates). (**B**) Upper: Sequence comparison of upstream regions of the five regulated genes. Putative consensus regions are shown in red and fully conserved nucleotides are shown in bold letter. Underlines represent transcribed regions based on the report (*Kopf et al., 2014b*) and initiation codons of regulated genes are shown in black/bold letters. (**B**) Lower: Sequences of DNA probe and competitors for *xssE* (native and mutant) used for electrophoretic mobility shift assay (EMSA) of C. Consensus regions are shown in red, and mutated regions are shown in blue. Total DNA size is 251 bp, where identical sequences are mostly not shown except 20 bp at both ends. (**C**) The autoradiogram image of EMSA of XssQ protein and the DNA probe of *xssE* with some competitors.

The online version of this article includes the following source data and figure supplement(s) for figure 3:

Source data 1. Raw data of RT-qPCR analysis in wild type (WT), Δ*xssS*, and Δ*xssQ*.

Figure supplement 1. Scatter plot of transcriptome comparison among wild type (WT), Δ*xssS*, and Δ*xssQ*.

Figure supplement 1—source data 1. RPKM (reads per kilobase of exon per million mapped reads) values of RNA-seq analysis of *Synechocystis* sp.

Figure supplement 2. Chromatograms of HPLC and anion exchange column chromatography of exopolysaccharides (EPS).

analyze the transcriptome (*Figure 3—figure supplement 1*). The genes down-regulated in Δ*xssQ* and up-regulated in Δ*xssS* were mostly *xss* genes. In detail, the regulated genes were *xssA-E* and *xssL-P*, which were roughly consistent with the qPCR analysis (*Figure 3A*). We conclude that *xssA-E* and *xssL-P* were specifically regulated by XssS/XsrR/XssQ. In a previous report, *xssA–xssE* and *xssL–xssP* were up-regulated at low temperature in another substrain of *Synechocystis* 6803 (*Kopf et al., 2014b*). To test this in our substrain, we measured the sulfated EPS accumulation of WT culture at 20°C, and it was 3.1-fold greater than that at normal growth temperature (31°C; *Figure 2D,E* and *Supplementary file 2*). This result suggests that XssS/XsrR/XssQ is a temperature sensor for *xss* gene expression.

We aligned nucleotide sequences near the transcription start sites of the regulated genes (*xssA, xssE, xssL, xssN*, and *xssP*) to find the consensus sequences for XssQ binding (*Figure 3B*), according to the differential RNA-seq-type transcriptomic analysis of Synechocystis 6803 (*Kopf et al., 2014b*). There are single or tandem consensus sequence, AAGTTXXAC. To confirm the binding of XssQ to this region, we performed electrophoretic mobility shift assay (EMSA) using purified recombinant XssQ protein and a PCR-amplified DNA fragment of *xssE* upstream (*Figure 3C*). The band position of the radiolabeled probe DNA shifted reflecting the concentration of XssQ. This shift was largely eliminated by excess addition of the unlabeled native competitor, but not by addition of the mutant competitor with mutations in the consensus region. These results suggest that XssQ recognizes the consensus sequence of *xssE* and other target genes for their transcriptional activation.

## The chemical composition of the sulfated EPS

The EPS fractions of WT and the overproduction mutant (Δ*xssS*) were subjected to chemical composition analysis (*Table 1*, *Figure 3—figure supplement 2*). EPS from Δ*xssS* consisted of only four types of monosaccharides and sulfate with the near stoichiometric molar ratio of rhamnose:mannose:galactose:glucose:sulfate of 1:1:1:5:2. This finding roughly fits with the gene number, that is, eight glycosyltransferase genes and two sulfotransferase genes. We assumed that the sulfated EPS of Δ*xssS* is produced by the *xss* cluster in Synechocystis 6803 and designated 'synechan'. On the other hand, the EPS fraction from WT likely contained unknown polysaccharides consisting of mannose, fucose, and xylose in addition to synechan of the Δ*xssS*. It should be mentioned that the membrane filtration is effective to collect synechan selectively from the 'free' polysaccharides, which have been mixed together with viscous molecules in literature.

## The OPX protein for synechan biosynthesis

There is no candidate gene in the *xss*-carrying plasmid for the OPX protein of the Wzx/Wzy system, whereas *sll1581*, an OPX homolog, was found on the main chromosome. Disruption of *sll1581* (Δ*sll1581*) abolished bloom formation and EPS accumulation (*Figure 2D,E*). Thus, the chromosomal OPX protein Sll1581 (XssT) appears to serve as the outer-membrane exporter for synechan. Interestingly, Synechocystis 6803 possesses *xssT* (OPX gene) and *sll0923* (a second PCP-2a gene) on the main chromosome and *xssK* (PCP-2a gene) on the plasmid pSYSM, whereas its close relative Synechocystis 6714 harbors only homologs of *sll0923* and *xssT* but lacks the entire plasmid carrying the *xss* cluster. This suggested that XssT serves as an OPX for dual function for both XssK and Sll0923. It is likely that Synechocystis 6803 acquired pSYSM and borrowed the chromosomal OPX gene *xssT* to produce synechan or, alternatively, Synechocystis 6714 may have lost the plasmid.

**Table 1.** Chemical composition of the exopolysaccharides (EPS) from wild-type (WT) Synechocystis 6803 and Δ*xssS* mutant.

| | Sugars | | | | | | | | | | | Sulfate residues |
|---|---|---|---|---|---|---|---|---|---|---|---|---|
| | Neutral sugars (mol/mol% ) | | | | | | | | Uronic acids (mol/mol % ) | | | Substitution degree |
| | Rhamnose | Ribose | Mannose | Fucose | Galactose | Xylose | Glucose | Total | Galacturonic acid | Glucuronic acid | Total | (mol/mol%) |
| WT | 16.6 | N.D. | 25.7 | 16.2 | 4.7 | 10.6 | 23.1 | 96.9 | ND | 3.1 | 3.1 | 10.4 |
| Δ*xssS* | 13.1 | N.D. | 14.2 | 1.2 | 12.5 | 1.0 | 57.9 | 99.9 | ND | 0.1 | 0.1 | 26.6 |

## Discussion

Summarizing these data, we propose models for synechan biosynthesis apparatus including OPX and temperature-responsive regulation (*Figure 4A,B* and *Figure 2—figure supplement 2*). The model of the Xss apparatus fits well with the known Wzx/Wzy-dependent apparatus represented by xanthan biosynthesis in *Xanthomonas campestris* (*Katzen et al., 1998*). The eight glycosyltransferases including XssP (the priming glycosyltransferase) produce oligosaccharide repeat unit of eight sugars, which is consistent with the sugar composition of synechan. These findings suggest that the *xss* cluster on the pSYSM plasmid harbors a whole set of genes for synechan biosynthesis except the OPX gene (*xssT* on the main chromosome). Notably, the cluster harbors two sulfotransferase genes, which have not been found to our knowledge in other bacterial gene clusters for extracellular

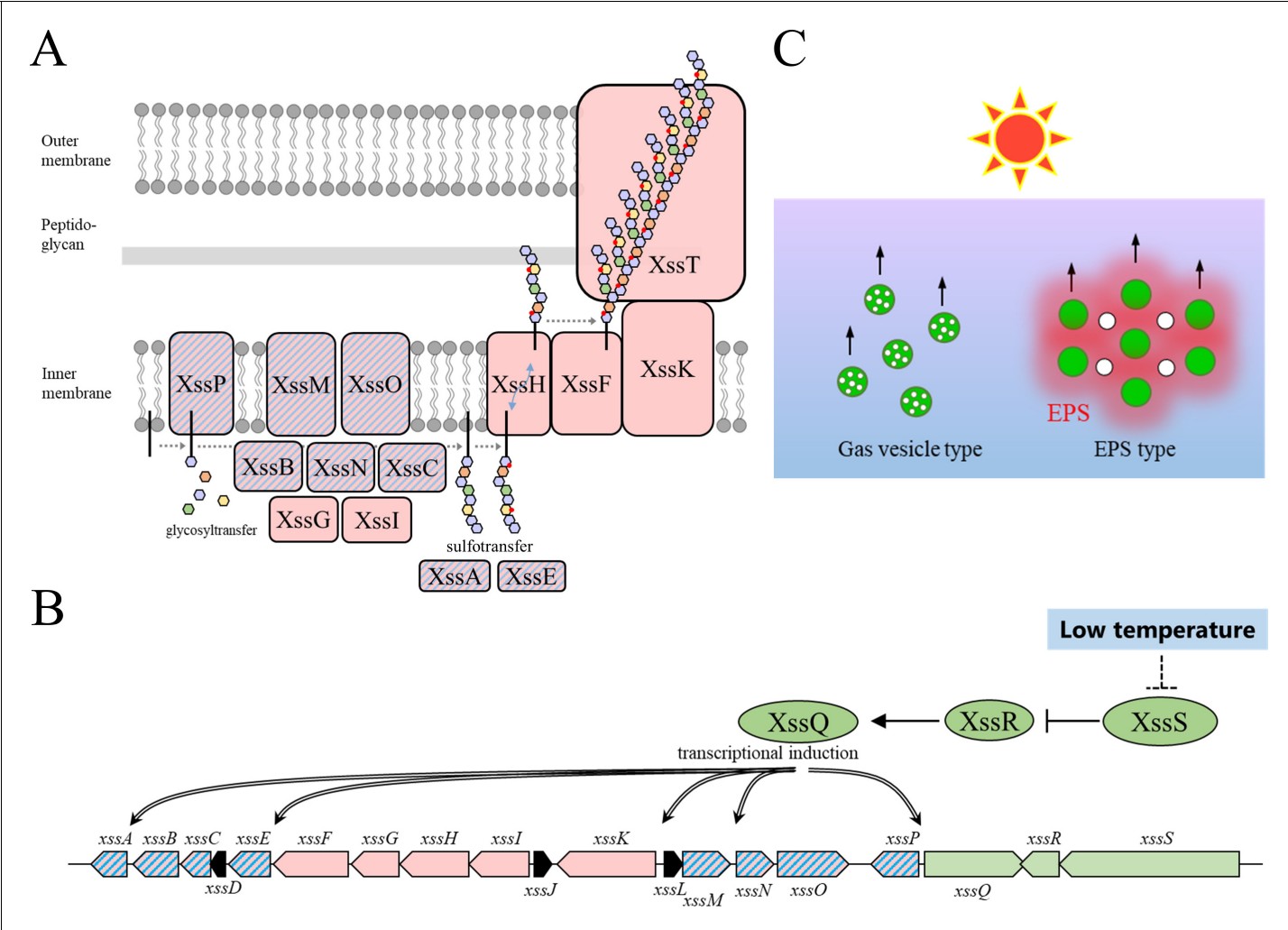

**Figure 4.** Proposed models for the synechan biosynthesis apparatus, transcriptional regulation, and bloom formation. (**A**) Model for the synechan biosynthesis apparatus with sugar polymerization and modification. Red boxes represent biosynthesis components, and red boxes with blue stripes represent transcriptionally regulated components. A putative lipid linker is shown as a black rod. Each monosaccharide is shown as a small hexagon, and each sulfate group is shown as a red spot. (**B**) Signaling and transcriptional regulation model. Green arrows and ellipses represent regulatory genes and proteins, respectively. Genes for synechan biosynthesis are shown in red, and transcriptionally regulated genes are depicted with blue stripes. Arrows with double lines represent transcriptional activation. (**C**) Two flotation models in cyanobacteria. Left, flotation of cells (green circles) with intracellular gas-filled vesicles (white circles). Right, flotation of exopolysaccharides (EPS) (red shading)-entrapped cells (green circles) and extracellular gas bubbles (white circles), which are generated by photosynthesis.

The online version of this article includes the following figure supplement(s) for figure 4:

**Figure supplement 1.** Typical examples of putative gene clusters for biosynthesis of sulfated polysaccharides in cyanobacteria.

**Figure supplement 2.** Homologous gene clusters of *xssS/xssR/xssQ* in some cyanobacteria.

polysaccharide biosynthesis. Sulfotransferases, XssA and XssE, belong to distinct subfamilies of bacterial sulfotransferases. We found many sulfotransferase genes in various cyanobacterial genomes by Pfam search (PF00685, PF03567, PF13469). They are mostly found in gene clusters for putative extracellular polysaccharide biosynthesis (Wzx/Wzy-type and ABC-type) (*Figure 4—figure supplement 1*). It should be noted that they are more or less partial as a cluster for extracellular polysaccharide biosynthesis system, whereas the *xss* cluster appears to be complete except the OPX gene in Synechocystis 6803. It is well established that the polysaccharide moiety of membrane-anchored lipopolysaccharides and CPS of bacteria are produced and exported by the Wzx/Wzy-dependent or ABC transporter–dependent pathways, whereas free EPS, that is, xanthan and cellulose, are produced by the Wzx/Wzy-dependent and synthase-dependent pathways but not by the ABC transporter–dependent pathway (*Schmid et al., 2015*; *Willis and Whitfield, 2013*). In the literature, a sulfated CPS was reported in *A. platensis* (formerly *Spirulina platensis*) (*Mouhim et al., 1993*). This sulfated CPS may be produced by an ABC transporter-type gene cluster in *Figure 4—figure supplement 1*. Gene disruption will confirm such predictions deduced from the gene cluster analyses, although targeted disruption is not so easy in many cyanobacteria due to poor transformation efficiency except Synechocystis 6803. In contrast, no sulfated polysaccharide has been reported in the other bacteria, though many sulfotransferases are also registered in Pfam database. Some of them are known to transfer a sulfuryl group to lipo-oligosaccharides in rhizobia (Nod factor) and mycobacteria (*Mougous et al., 2002*).

XssQ, a STAND protein with a DNA binding domain is indeed the transcriptional activator for *xssE* and other induced genes. XssQ homologs are found widely throughout the cyanobacteria but the set of XssS/XssR/XssQ is found near the gene cluster for sulfated EPS biosynthesis with sulfotransferases in many cyanobacteria (*Figure 4—figure supplement 2*, *Supplementary file 3*). Consensus sequences are also found in upstream of some genes in the cluster, suggesting that the XssS/XssR/XssQ system may operate universally for induction of sulfated EPS production under certain environmental conditions such as cold temperature.

Acidic polysaccharides containing uronic acids and other carboxylic groups are common in bacteria, but sulfated polysaccharides are produced exclusively by cyanobacteria (*Pereira et al., 2009*). To speculate on the physiological significance of sulfated polysaccharides, we summarized the distribution of sulfotransferase genes in cyanobacteria of various habitats (*Supplementary file 3*). Species living in high salinity environments such as a salt lake or seawater mostly produce sulfated polysaccharides. Abundant sulfotransferases are found in the marine Trichodesmium that forms bloom-like colonies to collect dust-bound iron for nutrition (*Rubin et al., 2011*). Sulfated polysaccharides may play a role for adsorbing metal ions such as iron in saline environments. Consistently, the *xss* genes are up-regulated in iron deficient conditions in Synechocystis 6803 (*Kopf et al., 2014b*). The freshwater species Synechocystis sp. PCC 6714 lacks the *xss* genes including sulfotransferases and salt-resistance genes that are present in the more salt-resistant Synechocystis 6803 (*Kopf et al., 2014a*). A cyanobacterial sulfated polysaccharide, sacran, shows much higher capacity for saline absorption than does uronic acid-containing polysaccharides, whereas both absorb pure water efficiently (*Okajima et al., 2008*). This fact may also support the role of sulfated polysaccharides for water retention in aquatic or terrestrial habitats.

There are many putative genes that may be involved in biosynthesis and export of extracellular polysaccharides and lipopolysaccharides in the genome of Synechocystis 6803 (*Fisher et al., 2013*; *Pereira et al., 2015*). Many of them have been disrupted for characterization. With regard to the Wzx/Wzy-dependent system, *sll5052* (*xssK*) disruption mutant showed no clear phenotypes for bloom formation or EPS production (*Jittawuttipoka et al., 2013*), probably because the parent strain did not produce discernable amount of EPS like our non-motile strain or large EPS molecules were removed by membrane filtration from their EPS preparation. Similarly, the deletion mutant of *sll5049* (*xssH*) did not show any defect in EPS or CPS accumulation, though related mutants (Δ*sll0923* for a second PCP-2a) were shown to be depleted slightly of both CPS and EPS (*Pereira et al., 2019*). These results contrast with our null phenotype of Δ*sll5052* (*xssK*) and Δ*sll5049* (*xssH*), probably because of the difference in the parent strains. In addition, we found that our Δ*sll0923* did not show any defect in the bloom formation. On the other hand, disruption of *sigF* (*slr1564*) for a sigma factor of global cell surface regulation increased three- to fourfold accumulation of sulfated EPS preparation (*Flores et al., 2019a*; *Flores et al., 2019b*). The proteome analysis of Δ*sigF* revealed many (more than 160) proteins except for any Xss proteins were up-regulated,

leaving the sulfated EPS biosynthesis pathway elusive. The sugar composition of the EPS of their WT is somehow similar to our WT, likely reflecting varied mixture of synechan and unknown polysaccharides. Moreover, the sugar composition of the EPS fraction of ΔsigF was different for WT or synechan from our ΔxssS. The global cell surface regulator *sigF* may affect accumulation of various polysaccharides.

To get insights into the difference in bloom formation between the motile and non-motile substrains, we compared transcriptome data (*Supplementary file 4*). It is evident that many *xss* genes on the plasmid are expressed several times higher in the motile substrain than the non-motile one except for *xssT* on the main chromosome, despite that the nucleotide sequence of the *xss* gene cluster was completely conserved between them. This fact suggests a possibility that another signaling via XssS/XssR/XssQ may contribute to the difference in bloom formation and synechan production between the substrains. Moreover, cell aggregation of another motile substrain of Synechocystis 6803 requires type IV pili apparatus, which drives cell motility (*Conradi et al., 2019*). Cell aggregation of another non-motile substrain also requires the type IV pili but this aggregation was enhanced by disruption of the OPX gene (*xssT*) (*Allen et al., 2019*). Thus, cell aggregation and bloom formation may be complex phenomena of extracellular polysaccharides and type IV pili in Synechocystis 6803.

The cyanobacterial bloom rapidly accumulates in populations of cyanobacterial cells floating on the water surface, which often produce potent cyanotoxins (hepatotoxins, neurotoxins, etc.) (*Merel et al., 2013*). Blooms are thought to be supported mainly by cellular buoyancy due to intracellular proteinaceous gas vesicles constructed by gas vesicle proteins (*Beard et al., 2002*; *Walsby, 1994*). Moreover, recent studies suggested that extracellular polysaccharides are also important for the bloom formation (*Chen et al., 2019*). Some papers reported that the cells without gas vesicle can form blooms by EPS-dependent manner after artificial addition of divalent cations ($Ca^{2+}$ or $Mg^{2+}$) (*Dervaux et al., 2015*; *Wei et al., 2019*). On the other hand, our study demonstrated that the gas-trapped EPS is sufficient for bloom formation of Synechocystis 6803, which does not produce gas vesicles, without addition of any divalent cations (*Figure 4B*). This result is consistent with reports that cyanobacteria without gas vesicles form booms in natural environments, including freshwater lakes (*Casero et al., 2019*; *du Plooy et al., 2015*; *Steffen et al., 2012*).

Finally, sulfated polysaccharides are expected to be healthy foods, industrial materials and medicines (*Jiao et al., 2011*; *Wardrop and Keeling, 2008*). Some sulfated EPS from Synechocystis 6803 showed antitumor activity (*Flores et al., 2019a*), though this EPS may not be identical to synechan. The Xss-dependent biosynthesis of synechan in Synechocystis 6803 should be a good model for studies of other cyanobacterial sulfated polysaccharides. Combinatorial expression of sulfotransferases and glycosyltransferases from other cyanobacteria in Synechocystis cells will provide clues to their functions. Heterologous expression of Synechocystis *xss* genes in other organisms will also open a possibility of large-scale production of modified synechan species. Further molecular studies of *xss* genes and related genes from the database should accelerate screening and potential applications of cyanobacterial sulfated polysaccharides.

## Materials and methods

**Key resources table**

| Reagent type (species) or resource | Designation | Source or reference | Identifiers | Additional information |
|---|---|---|---|---|
| Gene (Synechocystis sp. PCC 6803) | *slr1943* | GenBank | Gene ID:952818 | |
| Gene (Synechocystis sp. PCC 6803) | *slr1043* | GenBank | Gene ID:953647 | |
| Gene (Synechocystis sp. PCC 6803) | *sll0501* | GenBank | Gene ID:953286 | |

*Continued on next page*

Continued

| Reagent type (species) or resource | Designation | Source or reference | Identifiers | Additional information |
|---|---|---|---|---|
| Gene (Synechocystis sp. PCC 6803) | slr1118 | GenBank | Gene ID:952865 | |
| Gene (Synechocystis sp. PCC 6803) | sll5042 | GenBank | Gene ID:2655985 | |
| Gene (Synechocystis sp. PCC 6803) | sll5043 | GenBank | Gene ID:2655983 | |
| Gene (Synechocystis sp. PCC 6803) | sll5044 | GenBank | Gene ID:2655981 | |
| Gene (Synechocystis sp. PCC 6803) | ssl5045 | GenBank | Gene ID:2655897 | |
| Gene (Synechocystis sp. PCC 6803) | sll5046 | GenBank | Gene ID:2655982 | |
| Gene (Synechocystis sp. PCC 6803) | sll5047 | GenBank | Gene ID:2655980 | |
| Gene (Synechocystis sp. PCC 6803) | sll5048 | GenBank | Gene ID:2655974 | |
| Gene (Synechocystis sp. PCC 6803) | sll5049 | GenBank | Gene ID:2655975 | |
| Gene (Synechocystis sp. PCC 6803) | sll5050 | GenBank | Gene ID:2655972 | |
| Gene (Synechocystis sp. PCC 6803) | slr5051 | GenBank | Gene ID:2655936 | |
| Gene (Synechocystis sp. PCC 6803) | sll5052 | GenBank | Gene ID:2655973 | |
| Gene (Synechocystis sp. PCC 6803) | slr5053 | GenBank | Gene ID:2655990 | |
| Gene (Synechocystis sp. PCC 6803) | slr5054 | GenBank | Gene ID:2655991 | |
| Gene (Synechocystis sp. PCC 6803) | slr5055 | GenBank | Gene ID:2655867 | |
| Gene (Synechocystis sp. PCC 6803) | slr5056 | GenBank | Gene ID:2655868 | |
| Gene (Synechocystis sp. PCC 6803) | sll5057 | GenBank | Gene ID:2655970 | |
| Gene (Synechocystis sp. PCC 6803) | slr5058 | GenBank | Gene ID:2655931 | |

*Continued*

| Reagent type (species) or resource | Designation | Source or reference | Identifiers | Additional information |
|---|---|---|---|---|
| Gene (Synechocystis sp. PCC 6803) | *sll5059* | GenBank | Gene ID:2655971 | |
| Gene (Synechocystis sp. PCC 6803) | *sll5060* | GenBank | Gene ID:2655968 | |
| Gene (Synechocystis sp. PCC 6803) | *sll1581* | GenBank | Gene ID:953845 | |
| Strain, strain background (Synechocystis sp. PCC 6803) | Wild-type strain; WT | Doi.org/10.1093/dnares/dsr042 | PCC-P | Motile |
| Strain, strain background (Synechocystis sp. PCC 6803) | Non-motile strain; NM | Doi.org/10.1093/dnares/dsr042 | GT-I | Glucose-tolerant |
| Strain, strain background (*Escherichia coli*) | JM109 | Takara | Takara:9052 | |
| Recombinant DNA reagent | pPCR-Script (plasmid) | STRATAGENE | STRATAGENE:211186 | |
| Commercial assay or kit | In-Fusion HD Cloning | Clontech | Clontech:639635 | |
| Chemical compound, drug | Alcian Blue 8GX | MERCK | MERCK:05500 | |

## Cyanobacterial strains and cultures

The motile substrain PCC-P of the unicellular cyanobacterium Synechocystis sp. PCC 6803, which exhibits phototaxis (*Yoshihara et al., 2000*) and forms bloom-like aggregates, was used as the WT in this work. A non-motile glucose-tolerant substrain, which has been widely used for studies of photosynthesis, was used for comparison (*Chin et al., 2018*). Cells were maintained in 50 mL of BG11 liquid medium (*Stanier et al., 1971*) under continuous illumination of fluorescent lamps from outside (30 µmol photons/m$^2$/s) with bubbling of 1% $CO_2$ in air at 31℃, or on 1.5% agar plates. Cell density was monitored at 730 nm.

## Construction of plasmids and mutants

Primers used are listed in *Supplementary file 5*. Plasmids and mutants were constructed as described (*Chin et al., 2018*). In brief, the DNA fragments, antibiotic-resistance cassettes, the *trc* promoter, and plasmid vectors were amplified by PCR using PrimeSTAR MAX DNA polymerase (Takara, Shiga, Japan) and combined using the In-Fusion System (Takara). The resulting plasmid constructs were confirmed by DNA sequencing.

Gene disruption was performed in two different ways. One method was replacement of a large portion of a targeted gene(s) with an antibiotic-resistance cassette. The other method was replacement of the translation initiation codon with a stop codon. In both cases, the screening cassette without the terminator was inserted in the direction of the targeted gene(s) to allow transcriptional read-through of the downstream gene(s). For overexpression, gene expression was constitutively driven by the strong *trc* promoter in two ways: integration of a target gene with the strong *trc* promoter into a neutral site near *slr0846* or IS203c, or replacement of the target-gene promoter with the *trc* promoter. Natural transformation and subsequent homologous recombination were performed as described (*Chin et al., 2018*). The antibiotic concentration for the selection of transformants was 20 µg/mL chloramphenicol, 20 µg/mL kanamycin, and/or 20 µg/mL spectinomycin. Complete segregation of the transformed DNA in the multicopy genome was confirmed by PCR

using primers listed in *Supplementary file 5*, and the transformants are listed in *Supplementary file 6*.

## Bloom formation

The bloom was reproducibly formed using the two-step culture regime we developed in this work. Before the bloom formation experiment, cells were precultured once in liquid after transfer from plates. In the first step, cells inoculated at $OD_{730} = 0.2$ were grown with vigorous aeration under continuous light at 31℃ or 20℃ for 48 hr. Typically the cell density reached $OD_{730} \sim 2$. In the second step, the culture was shifted to the standing condition without bubbling under the same continuous light for another 48 hr (or longer) for cells to rise to the surface. Regarding the mutants of transmembrane glycosyltransferases, bloom formation was examined after 168 hr of the second-step culture. The final concentration of the photosynthesis inhibitor DCMU (3-(3,4-*di*chlorophenyl)-1,1-di*methy*l*u*rea) was 100 µM.

## EPS fractionation

The fractionation method to isolate the crude EPS is shown in *Figure 1—figure supplement 1A*. The viscous materials including cells after the second step of culture were collected by filtration using a 1.0 µm pore PTFE membrane (Millipore). The trapped materials were gently and carefully recovered from the membrane using MilliQ water with the aid of flat-tip tweezers. The collected sample was vortexed and then centrifuged at $20,000 \times g$ for 10 min to remove cells. The supernatant constituted the crude EPS that contained viscous EPS and possibly CPS.

The refined fractionation method to isolate EPS is shown in *Figure 1F*. The entire culture at the end of the first step, which did not contain gas bubbles, was first centrifuged at $10,000 \times g$ for 10 min to remove cells and CPS and then filtered through a 1.0 µm pore PTFE membrane. The trapped EPS was carefully recovered as described above. The flowthrough of the filtration was regarded as free polysaccharides, which were recovered by ethanol precipitation. CPS was released from the cell pellet by vigorous vortexing with MilliQ water and recovered by centrifugation to remove cells ($20,000 \times g$ for 10 min).

## Sugar quantification

Total sugar was quantified using the phenol-sulfate method (*DuBois et al., 1956*). A 100 µL aliquot of 5% (w/w) phenol was added to 100 µL of a sample in a glass tube and vortexed three times for 10 s. Then, 500 µL of concentrated sulfuric acid was added, and the tube was immediately vortexed three times for 10 s and then kept at 30℃ for 30 min in a water bath. Sugar content was measured by absorption at 487 nm using a UV-2600PC spectrophotometer (Shimadzu, Japan, Tokyo). Any contamination of the BG11 medium was evident by slight background coloration. This background was subtracted on the basis of the extrapolation of absorption at 430 nm, where the coloration due to sugars was minimal. Glucose was used as the standard. Some EPS samples were highly viscous, so we vortexed and sonicated them before measurement. Statistical significance was determined using Welch's *t* test.

## Sugar composition analysis

The collected EPS samples were dialyzed with MilliQ water and then freeze-dried for 3 days. Sugar composition was analyzed by Toray Research Center, Inc (Tokyo, Japan). A part of the fluffy sample (WT, 0.298 mg; Δ*xssS*, 0.203 mg) was dissolved in 200 µL of 2 M trifluoroacetic acid and hydrolyzed at 100℃ for 6 hr. The treated sample was vacuum-dried with a centrifugal evaporator, redissolved in 400 µL MilliQ water, and filtered through a 0.22 µm pore filter. This sample was used for the analysis.

Monosaccharide composition was determined by HPLC with the LC-20A system (Shimadzu). For neutral sugars, the column was TSK-gel Sugar AXG (TOSOH, Japan) and the temperature was 70℃. The mobile phase was 0.5 M potassium borate (pH 8.7) at 0.4 mL/min. Post-column labeling was performed using 1% (w/v) arginine and 3% (w/v) boric acid at 0.5 mL/min, 150℃. For uronic acids, the column was a Shimpack ISA-07 (Shimadzu) and the temperature was 70℃. The mobile phase was 1.0 M potassium borate (pH 8.7) at 0.8 mL/min. Post-column labeling was performed using 1% (w/v) arginine and 3% (w/v) boric acid at 0.8 mL/min, 150℃. The detector was an $RF-10A_{XL}$

(Shimadzu), with excitation at 320 nm and emission at 430 nm. The standard curves were prepared for each monosaccharide with standard samples.

The $SO_4^{2-}$ content was determined by anion exchange column chromatography using the ISC-2100 system (Thermo Fisher Scientific, Waltham, MA). The column was eluted via a gradient of 0–1.0 M KOH. The separation column was IonPac ASI l-HC-4 µm (Thermo Fisher Scientific). Electric conductivity was used for detection.

### Alcian blue staining

The polysaccharides were stained with 1% Alcian blue 8GX (Merck) for 10 min in 3% acetic acid (pH 2.5) or 0.5 N HCl (pH 0.5) as previously described (*Di Pippo et al., 2013*).

### Quantitative PCR

The qPCR was performed as described in our previous work (*Maeda et al., 2018*). Cells were harvested by centrifugation at 5000 × *g* for 10 min at 4°C. Cell disruption and RNA extraction were done using an RNeasy Mini kit for bacteria (Qiagen, Venlo, The Netherlands). In addition, cells were disrupted five times by mechanical homogenization with zirconia beads (0.1 mm diameter) in a microhomogenizing system (Micro Smash MS-100, TOMY SEIKO, Tokyo, Japan) at 5000 rpm for 40 s. For cDNA preparation, RNA was reverse-transcribed using random primers (PrimeScript RT reagent kit with gDNA eraser, Takara). Real-time PCR was performed using the THUNDERBIRD SYBR qPCR Mix (Toyobo) and the Thermal Cycler Dice Real Time System II (Takara). The transcript level in each strain was normalized to the internal control (*rnpB*). The primers used are listed in *Supplementary file 5*.

### Electrophoretic mobility shift assay

The expression and purification of recombinant His-tagged proteins and EMSA were performed as described in our previous works (*Hirose et al., 2010*; *Maeda et al., 2014*). In brief, His-tagged XssQ was expressed using pET28a vector system and *Escherichia coli* C41(DE3) strain. The protein was purified by Histrap HP column (Cytiva, Tokyo, Japan) and AKTA prime system (Cytiva). For probe and native competitor, the upstream region of *xssE* was amplified with the primer set xssEup-1F/2R (total 251 bp). As a mutant competitor, the same region of the chemically synthesized DNA fragment containing mutations in the two consensus sequences was used for amplification with the same primer set as mentioned above. Labeling of the DNA probe, electrophoresis, and autoradiography were performed as described (*Midorikawa et al., 2009*). We incubated the aliquots of the XssQ protein (0, 500, 1000, or 3000 ng/lane) with the radiolabeled probe for 30 min at room temperature. For competition, 3000 ng of XssQ was incubated with the probe and 20 pmol of unlabeled competitors (native or mutant).

### RNA-seq analysis

RNAs for RNA-seq analysis were extracted as described above. Library construction for RNA sequencing analysis was conducted as described previously (*Ohbayashi et al., 2016*). The average number of raw read pairs per sample was 2.84 million. The reads were trimmed using CLC Genomics Workbench ver. 12.0 (QIAGEN) with the following parameters; Phred quality score >30; ambiguous nucleotides allowed: 1; automatic read-through adaptor trimming: yes; removing the terminal 15 nucleotides from the 5′ end; and removing truncated reads of less than 20 nucleotides in length. Trimmed reads were mapped to the all genes in Synechocystis sp. PCC 6803 using CLC Genomics Workbench ver. 12.0 with the following parameters; mismatch cost: 2; indel cost: 3; length fraction: 0.8; similarity fraction: 0.9; and maximum number of hits for a read: 10. In the comparison between WT and mutants ΔxssS, and ΔxssQ (*Figure 3—figure supplement 1*), the genome information (accession numbers, chromosome: CP003265, pSYSM: CP003266, pSYSX: CP003269, pSYSA: CP003267, pSYSG: CP003268, pCA2.4: CP003270, pCB2.4: CP003271, and pCC5.2: CP003272) was used as a reference, and in the comparison between the motile WT (PCC-P) and non-motile (NM) substrains (*Supplementary file 4*), the genome information (accession numbers, chromosome: AP012276, pSYSM: AP004310, pSYSX: AP006585, pSYSA: AP004311, pSYSG: AP004312, pCA2.4: CP003270, pCB2.4: CP003271, and pCC5.2: CP003272) was used as a reference. Reads per kilobase per million mapped reads (RPKM) were calculated using CLC Genomics Workbench ver. 20.0.

Original sequence reads were deposited in the DRA/SRA database with the following accession numbers, DRA011755. The accession number of BioProject was PRJDB11449.

## Bioinformatics analysis

The sequences of the proteins were obtained from NCBI (http://www.ncbi.nlm.nih.gov/) and Pfam (http://pfam.xfam.org/) (*Finn et al., 2016*). The domain architecture was searched using the Simple Modular Architecture Research Tool, SMART (*Letunic et al., 2015*). Glycosyltransferase classifications were based on the CAZy database (http://www.cazy.org/) (*Henrissat, 1991*; *Lombard et al., 2014*). Amino acid sequence similarity was evaluated by NCBI BLAST search.

## Acknowledgements

This study was supported by Grants-in-Aid for JSPS Fellows 15J07605 and 19J01251 (to KM), the Japan Society for the Promotion of Science for Scientific Research (16H06558 to MI), and the JST for CREST program (JPMJCR1653 to MI).

## Additional information

### Funding

| Funder | Grant reference number | Author |
| --- | --- | --- |
| Japan Society for the Promotion of Science | 15J07605 19J01251 | Kaisei Maeda |
| Japan Society for the Promotion of Science | 16H06558 | Masahiko Ikeuchi |
| Japan Science and Technology Agency | JPMJCR1653 | Masahiko Ikeuchi |

The funders had no role in study design, data collection and interpretation, or the decision to submit the work for publication.

### Author contributions

Kaisei Maeda, Conceptualization, Resources, Data curation, Formal analysis, Supervision, Funding acquisition, Investigation, Visualization, Methodology, Writing - original draft, Project administration, Writing - review and editing; Yukiko Okuda, Resources, Investigation, Methodology; Gen Enomoto, Supervision, Investigation; Satoru Watanabe, Resources, Data curation, Supervision, Investigation, Methodology; Masahiko Ikeuchi, Conceptualization, Resources, Supervision, Funding acquisition, Validation, Writing - original draft, Project administration, Writing - review and editing

### Author ORCIDs

Masahiko Ikeuchi (iD) https://orcid.org/0000-0003-4231-8423

### Decision letter and Author response

Decision letter https://doi.org/10.7554/eLife.66538.sa1
Author response https://doi.org/10.7554/eLife.66538.sa2

## Additional files

### Supplementary files

• Source data 1. Source data for *Supplementary file 4*.

• Supplementary file 1. Summary of Synechocystis 6803 Xss proteins. Function: GT, glycosyltransferase; ST, sulfotransferase; PCP, polysaccharide co-polymerase; OPX, outer-membrane polysaccharide export protein. Regulation: transcriptional regulation of *xss* genes by XssQ/R/S. n.d., not determined.

• Supplementary file 2. Exopolysaccharides (EPS) accumulation and bloom formation by wild-type (WT) Synechocystis 6803 and extracellular sulfated polysaccharide biosynthesis (*xss*) mutants. Classification: ST, sulfotransferase; GT, glycosyltransferase; PCP, polysaccharide co-polymerase; OPX, outer-membrane polysaccharide export protein. Total sugar content of the EPS fraction is expressed as µg glucose/1 × 10$^8$ cells. The errors are based on SD (n = 3). *Close to the detection limit. EPS accumulation ratio, percent of WT grown at 31°C. Bloom formation is summarized from *Figure 2*.

• Supplementary file 3. Number of sulfotransferase genes (STs) in the genome of cyanobacteria collected from various habitats. * mentioned in the text.

• Supplementary file 4. Transcriptional levels of extracellular sulfated polysaccharide biosynthesis (*xss*) genes in the motile (WT) and non-motile (NM) substrains. Transcriptional level is given as RPKM (reads per kilobase of exon per million mapped reads).

• Supplementary file 5. Primer sequences.

• Supplementary file 6. List of mutants. *Almost complete segregation. #Partial segregation.

• Transparent reporting form

### Data availability

All data generated or analysed during this study are included in the manuscript and supporting files. Source data files have been provided for Figure 1G, Figure 2C andE, Figure 3A, Figure 3-figure supplement 1 and Supplementary File 4.

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
