## [Decision Letter]

**Acceptance summary:**

With the findings in your article you have elucidated the biochemical and regulatory apparatus for the biosynthesis of sulfated exopolysaccharides, an entire class of molecules previously not studied in cyanobacteria. Therefore, the work has broad implications for the microbiology and ecology of these organisms but opens also the possibility to use these compounds in biotechnology and modify their structures by combinatorial synthesis in the future.

**Decision letter after peer review:**

Thank you for submitting your article "Biosynthesis system of Synechan, a sulfated exopolysaccharide, in the model cyanobacterium Synechocystis sp. PCC 6803" for consideration by *eLife*. Your article has been reviewed by 3 peer reviewers, including Wolfgang Hess as the Reviewing Editor and Reviewer #1, and the evaluation has been overseen by Gisela Storz as the Senior Editor. The following individuals involved in review of your submission have agreed to reveal their identity: Conrad W Mullineaux (Reviewer #2); Jochen Schmid (Reviewer #3).

Essential revisions:

The text of the manuscript needs to be revised to improve clarity in presentation and discussion.

*Reviewer #1 (Recommendations for the authors):*

The observations are very interesting and worthwhile to be published. Please consider the following points to improve the manuscript.

Line 21, 33-35: Some extracellular polysaccharides can be utilized for food, cosmetics, in medicine and as beneficial biomaterial, they have potential application in biotechnology. Can you give some real examples or be more specific instead of citing some reviews?

Line 97, 98, Figure S2: 5 genes encoding proteins with glycosyltransferase domains were screened by mutagenesis. I assume there are many more genes with such domains in Synechocystis. How many were bioinformatically screened initially? And which considerations led to the choice of those 5 genes for which deletions were constructed?

Line 160: The gene cluster on pSYSM contains also several "small proteins of unknown function". I noticed that xxsL/slr5053 has an annotation as "cyanoexosortase A system-associated protein".

Figure 3A: Y-axis, this should be "Relative transcript levels".

Line 234: Please add which Figure the statement refers to after "…which were roughly consistent with the qPCR analysis" (probably Figure 3A).

Line 235-236: "In a previous report, xssA-xssE and xssL-xssP were up-regulated at low temperature.…" Looking at the referred data I think the genes were upregulated at low temperature, by the shift to high light and iron starvation. This is especially well visible for the TSSs in front of slr5055, sll5057 and probably widens the relevance also in the ecological context. You may note that it was reported that natural populations of Trichodesmium with its record number of sulfotransferase genes according to your Table S3 form large floating colonies to collect iron-rich dust particles (https://www.nature.com/articles/ngeo1181 and PMID: 31551530).

Table 1: Measuring the chemical composition of the EPS found rhamnose to make up 16.6 mol/mol % of the pool of neutral sugars. The presence of rhamnose can be relevant for recently established approaches to use the rhamnose-inducible rhaBAD promoter in cyanobacteria (Kelly et al., 2018; PMID: 29544054). This promoter is considered strong and well-regulated and is becoming very popular. Especially, Synechocystis was not found to produce or metabolize rhamnose and therefore could be considered as an ideal inducer molecule. Can you comment on whether these ideas might need to be re-considered in view of your findings?

Transcriptome analysis: Please explain the legend to Figure S6 what the location of a dot in a particular quadrant means (upregulation in ΔxssS, downregulation in ΔxssS and ΔxssQ, etc.). It seems there were several genes down-regulated in ΔxssQ and down-regulated ΔxssQ. Can you please comment on what those genes are?

With the given information it is not possible to follow the RNA-seq analysis. Did you do replicates? How many reads were obtained? Which programs with which parameters were used for the QC, mapping etc.? This information cannot be inferred from the cited reference Ohbayashi et al., 2016. Please upload the raw reads to a public repository and include the accession number in the manuscript.

*Reviewer #2 (Recommendations for the authors):*

This is a very well-written and well-presented piece of work. The combination of genetic, chemical and phenotypic analysis is extremely thorough. My only significant issue with the paper is that the cartoon in Figure 4B implies that blooms are formed simply by embedding cells in EPS. This is clearly an over-simplification. The authors' data in Figure 2 and Table S2 shows that some mutants still have unaltered bloom formation despite a five-fold reduction in EPS production. Other recent work (not cited) has shown that the Type IV pili, and specific minor pilins, are essential for aggregation and floc formation in Synechocystis (AEM 85 (2019) e01292-18 ; J Bact 201 (2019) e00344-19). Especially the "flocs" discussed in the latter paper look just the same as the "blooms" discussed here. So I think the authors' discussion currently misses the chance to present a more complete and realistic model for bloom formation in Synechocystis. The obvious possibility is that Synechocystis uses its Type IV pili to anchor the cells to strands of EPS. That would explain why only small amounts of EPS are sufficient for full bloom formation.

*Reviewer #3 (Recommendations for the authors):*

The manuscript is of high interest for the scientific community and is in general scientifically and technically sound.

---

## [Author Response]

Reviewer #1 (Recommendations for the authors):The observations are very interesting and worthwhile to be published. Please consider the following points to improve the manuscript.Line 21, 33-35: Some extracellular polysaccharides can be utilized for food, cosmetics, in medicine and as beneficial biomaterial, they have potential application in biotechnology. Can you give some real examples or be more specific instead of citing some reviews?

We added several examples instead of general description, as suggested.

Line 97, 98, Figure S2: 5 genes encoding proteins with glycosyltransferase domains were screened by mutagenesis. I assume there are many more genes with such domains in Synechocystis. How many were bioinformatically screened initially? And which considerations led to the choice of those 5 genes for which deletions were constructed?

In the genome, 59 genes were annotated as glycosyltransferases. 12 of them were uncharacterized and predicted to encode transmembrane helices. In our initial work, we chose 5 genes. We added a short sentence to explain this process.

Line 160: The gene cluster on pSYSM contains also several "small proteins of unknown function". I noticed that xxsL/slr5053 has an annotation as "cyanoexosortase A system-associated protein".

Yes, we know the annotation. It is very intriguing that the xss cluster includes a exosortase-associated protein. But, the genome does not harbor any homologous gene for exosortase itself. We do not mention it in the current manuscript but would like to work in future.

Figure 3A: Y-axis, this should be "Relative transcript levels".

We corrected as suggested.

Line 234: Please add which Figure the statement refers to after "…which were roughly consistent with the qPCR analysis" (probably Figure 3A).

We added Figure 3A in the sentence as suggested.

Line 235-236: "In a previous report, xssA-xssE and xssL-xssP were up-regulated at low temperature.…" Looking at the referred data I think the genes were upregulated at low temperature, by the shift to high light and iron starvation. This is especially well visible for the TSSs in front of slr5055, sll5057 and probably widens the relevance also in the ecological context. You may note that it was reported that natural populations of Trichodesmium with its record number of sulfotransferase genes according to your Table S3 form large floating colonies to collect iron-rich dust particles (https://www.nature.com/articles/ngeo1181 and PMID: 31551530).

We thank for your helpful comments. We added discussion about iron adsorption and starvation from line 241 in the revised manuscript.

Table 1: Measuring the chemical composition of the EPS found rhamnose to make up 16.6 mol/mol % of the pool of neutral sugars. The presence of rhamnose can be relevant for recently established approaches to use the rhamnose-inducible rhaBAD promoter in cyanobacteria (Kelly et al., 2018; PMID: 29544054). This promoter is considered strong and well-regulated and is becoming very popular. Especially, Synechocystis was not found to produce or metabolize rhamnose and therefore could be considered as an ideal inducer molecule. Can you comment on whether these ideas might need to be re-considered in view of your findings?

Rhamnose is usually incorporated into polysaccharides by glycosyltransferase from dTDP-rhamnose, which is produced from dTDP-glucose in *Synechocystis* 6803. So, free rhamnose may be essentially absent in Synechocystis cells. We do not mention it in the manuscript because it is out of scope of the manuscript.

Transcriptome analysis: Please explain the legend to Figure S6 what the location of a dot in a particular quadrant means (upregulation in ΔxssS, downregulation in ΔxssS and ΔxssQ, etc.). It seems there were several genes down-regulated in ΔxssQ and down-regulated ΔxssQ. Can you please comment on what those genes are?

We added sentences for explanation of genes in the second quadrant, as suggested. Incidentally, we noticed that some genes of nitrogen assimilation (*nrtA-D*, *nirA* and *cynS*) are found in the fourth quadrant and genes for sulfate transport (*sbpA* etc.) are found in the third quadrant. But the exact DNA motif is not found in these genes. So, we did not include such description in the revised manuscript. Anyway, all the data are presented in the Excel file.

With the given information it is not possible to follow the RNA-seq analysis. Did you do replicates? How many reads were obtained? Which programs with which parameters were used for the QC, mapping etc.? This information cannot be inferred from the cited reference Ohbayashi et al., 2016. Please upload the raw reads to a public repository and include the accession number in the manuscript.

We added information in the Method and deposited the data in the public repository as accession number *DRA011755 in the revised manuscript*. Because RNA seq reads were large enough as mentioned in the Method, we hope our data are reliable although the experiment was done once.

Reviewer #2 (Recommendations for the authors):This is a very well-written and well-presented piece of work. The combination of genetic, chemical and phenotypic analysis is extremely thorough. My only significant issue with the paper is that the cartoon in Figure 4B implies that blooms are formed simply by embedding cells in EPS. This is clearly an over-simplification. The authors' data in Figure 2 and Table S2 shows that some mutants still have unaltered bloom formation despite a five-fold reduction in EPS production. Other recent work (not cited) has shown that the Type IV pili, and specific minor pilins, are essential for aggregation and floc formation in Synechocystis (AEM 85 (2019) e01292-18 ; J Bact 201 (2019) e00344-19). Especially the "flocs" discussed in the latter paper look just the same as the "blooms" discussed here. So I think the authors' discussion currently misses the chance to present a more complete and realistic model for bloom formation in Synechocystis. The obvious possibility is that Synechocystis uses its Type IV pili to anchor the cells to strands of EPS. That would explain why only small amounts of EPS are sufficient for full bloom formation.

We would like to focus on the flotation model in Figure 4C (not 4B), which only showed trapping of gasses within cells (gas vesicles) or EPS. However, we agree with the reviewer’s view that pili are also involved in bloom formation of this cyanobacterium. We discussed the role of pili in the Discussion with references as suggested (lines 274 to 280 in the revised manuscript). Anyway, the reference (Allen et al. 2019 AEM) reported that the cell aggregation was inhibited in the pili mutants but not in the OPX mutant. We believe more detailed study must be done for consensus of the cell aggregation mechanism. Regarding the apparent inconsistency between EPS and bloom of several mutants, we must be more careful to judge the bloom formation because it depends on time and other culture conditions. We would like to only mention the rough correlation between EPS accumulation and bloom formation in the current report.